# Compliance with Iron and Folic Acid Supplementation (IFAS) and associated factors among pregnant women in Sub-Saharan Africa: A systematic review and meta-analysis

Meseret Belete Fite [1]*, Kedir Teji Roba[2], Lemessa Oljira[2], Abera Kenay Tura[2], Tesfaye Assebe Yadeta [2]

1 Department of Public Health, Health Science Institute, Wollega University, Nekemte, Ethiopia,
2 Department of Public Health, School of Public Health, College of Health and Medical Sciences, Haramaya University, Harar, Ethiopia

☯ These authors contributed equally to this work.
* meseretphd2014@gmail.com

**Data Availability Statement:** All relevant data are within the paper and its Supporting Information files.

## Abstract

### Background

Anemia is one of the world's leading cause of disability and the most serious global public health issues. This systematic review and meta-analysis was conducted very carefully in order to give up the pooled compliance of Iron and Folic-Acid Supplementation in Sub-Saharan Africa.

### Methods

To conduct this brief systematic review and meta-analysis, a related literature search was done from different sources, PubMed Medline and Google Scholar Journals. Then IFA Supplementation related searching engine was used to make the work more meaningful and intensive. Moreover, we used modified Newcastle-Ottawa quality assessment scale for cross sectional studies to assess the quality of the study in terms of their inclusion. Then, the Preferred Reporting Items for Systematic Reviews and Meta-Analyses (PRISMA) guideline was followed to carry out the work in a carful manner. Finally, the pooled effect size was computed using the review manager and Compressive Meta-analysis software.

### Results

Twenty-three studies, which encompassed 24272 pregnant women, were chosen for the analysis. From those an overall prevalence of compliance with Iron and Folic Acid Supplementation (IFAS) in pregnancy in SSA was 39.2%. However, the result from meta-analysis showed that women who were counseled on IFAS in their courses of pregnancy were 1.96 times more likely to adhere to IFAS compared to those who were not counseled [OR:1.96, 95% CI (1.76-,5.93)]. Moreover, it showed that women who had knowledge of IFAS were

**Funding:** The authors received no specific funding for this work.

**Competing interests:** The authors have declared that no competing interests exist.

**Abbreviations:** ANC, Antenatal care; WHO, World Health Organization; IFAS, Iron and Folic-Acid Supplementation.

2.71 times more likely to have compliance with IFAS as compared to those who had no knowledge of IFAS [OR:2.71, 95% CI (1.33,5.54)]. Also it revealed that those women who had knowledge of anemia were 5.42 times more likely to have compliance with IFAS as compared with those who had no knowledge of anemia [OR5.42, 95% CI (1.52, 19.43)]. Furthermore, women who had received fourth visit for ANC were 1.54 times more likely to have compliance with IFAS as compared to those who had not received for ANC [OR 1.54, 95% CI (0.66, 3.58.43)].

## Conclusions

Our finding from this systematic review and meta-analysis shows the low case in prevalence of compliance to IFAS among pregnant women in SSA. Predictors for this includes: knowledge about anemia, knowledge about IFAS, counseling on IFAS and receiving fourth antenatal care visit were statistically correlated positively with compliance to IFAS. This demands careful appraisal of effect of prevention work for functioning policy, programs and plan nutrition intrusions for refining maternal dietary intake in gestation. Also dietary education intrusion requires to be planned to satisfy the needs of pregnant women. So we hope that the result of this study might be essential as a bridging stone for policy makers of Africa; exclusively for maternal and child health care. Finally, we recommended further studies to be conducted in the area of the study for more intensive and detailed suggestions.

## Introduction

In all over the world, anemia is one of the public health problems and continued as a universal top cause of frailty and the highest critical global health matters. This is because in a pregnancy, it is extremely predominant both in industrialized and unindustrialized countries. Recent evidence from World Health Organization (WHO) document revealed that, nearly 38% (32 million) women in pregnancy are anemic in the word. Out of this, 46.3% (9.2 Million) of them are in Africa [1]. However, the elucidations of the rate mostly show a discrepancy in the world from place to place. [2]. For instance, there is substantial deviation in the rate of anemia in pregnancy within advanced countries like USA in which the rate is 18%, in Australia 20%, in Singapore 67.8% and in china 70%; whereas the rates rises through trimesters [3–5]. But the magnitude of the rate is getting higher in unindustrialized nations; for example in Ethiopia 50.1%, in Sudan 53%, in Guinea 71% and in Pakistan 76.7%. These are the fundamental rationale problems related to anemia, which is one of the basic concerns of public health issues in the world in general and in Africa in particular [6–8].

Result from a number of studies conducted on this current issue showed that anemia in pregnancy has been connected to adverse pregnancy outcome and fetal growth [9]. This sound effect comprises: premature birth, LBW, abortion, delay psychomotor improvement, impairment of cognitive recital and reduce totals on intelligence (IQ) test level of the newly born baby which has an impact on the later life of the children at all [10–16]. Moreover, the influence of iron deficiency anemia (IDA) in first stages of teenager and early youthful are not possibly to be accustomed by substantial iron management [14]. This is because the iron dietary consumption upraises maternal mean hemoglobin concentration reads from 4.59 to 5.46g/L. Hence, excessive consumption of dietary iron at first or subsequent trimester pregnancy is meaningfully associated with decrement of the threat of anemia. This results in lessens

of adverse birth outcome, premature birth and LBW [6]. Likewise, women in Sub-Saharan Africa (SSA) intake low dietary iron, Calcium and Folic-Acid having less than RDA (Recommended Dietary Allowances) requirements for a woman during pregnancy for the reason that they were economically not recognized [15, 16].

Moreover, a plenty of works had examined multiple aspects upsetting anemia in pregnancy. For instance, independent predictors which include maternal age, residence, literateness, antenatal care visit, inter-pregnancy interval, iron food consumption, dietary practice, micronutrient intake, dietary diversity, iron supplementation, parasite infection and gravidity were documented as factors associated with developing anemia in pregnancy [17–19]. Moreover, women of third trimester pregnancy are more likely risky to develop anemia as compared to first and second trimester [20]. Therefore, WHO suggests day-to-day supplementation of 30–60 mg/d elemental iron (+400 $\mu$g) and folic acid to lessen the burden of anemia as a public health problem [1]. However, some other studies also reported that adherence to Iron and Folic-Acid Supplementation (IFAS) in Sub-Saharan Africa countries has a better position to some extent; however quiet leftovers at substandard level in which adherence proportion ranges from 10.6% in Kenya to79% in Mozambique [21, 22].

A number of recently published articles on compliance with IFAS in pregnancy in Sub-Saharan African counties are documented [23–45], but there is no systematic review and meta-analysis conducted on prevalence of compliance with IFAS and its determinants in SSA. Moreover, the current overall prevalence of compliance with IFAS in pregnancy is not well-known in this setup empirically. Therefore, the problem could be undetectable to policy makers. Thus, in order to sum up studies conducted in different corners of SSA countries and give overall prevalence of compliance with IFAS and its determinants; this systematic review and meta-analysis was conducted carefully to alleviate the problem.

## Methods

To conduct this brief systematic review and meta-analysis, a related literature of articles from PubMed, Medline and Google Scholar journal data base were collected. To enhance the chance of all-inclusiveness of the findings, uniterms and Bolen operators in English were used in searching strategies. Terms used for searching were: adherence OR compliance OR Iron and Folic-Acid Supplementation (IFAS) OR pregnancy OR pregnant women OR determinants OR factors OR Iron and Folic-Acid Supplementation (IFAS) and name of African countries. Finally, the results of this review were reported based on the Preferred Reporting Items for Systematic Review and Meta-Analysis statement (PRISMA) guideline.

### Selection of the studies

All articles related to prevalence and determinant of compliance with IFAS were collected from different sources. Since the year of publication for each articles were not limited, all articles published up to February 25, 2020 were incorporated for their eligibility in the review. Then, quantitative cross-sectional study design was used to make the work more clear and meaningful. However, articles published in qualitative methods were excluded because of the nature of the review and analysis chosen to be used in this paper. Then to have a deep understanding of each article, all authors read the title and abstract part independently. To avoid biases, all eligible articles were screened and selected after all individual's full reading of the abstract section of each article. Then the divergence of the work was managed to enhance the reliability and validity of the review and analysis based on pre-set inclusion criteria.

## Data extraction and quality assessment

In the process of making any review and analysis, the participation of more individuals has a significant value in data extraction. This is because of the principle of one plus one is greater than one. That is to mean scrutinized and condensed extracted data was obtained from more individual if they are participated equally on the extraction of a given raw data. To obey this rule and make the data more readable and meaningful, all authors were engaged in extraction. To do so data extraction template, which included author's name, year of publication, study location, sample size, odds ratio, confidence intervals and P-value, were prepared before the extraction of data was carried out. However, after the extraction of the data by each individual independently, we made a cross checked and compared the results very carefully. To make the work to address the target objective, all of us discussed and came to consensus on little partiality observed between us during the work. Thus, Modified Newcastle-Ottawa quality assessment scale for cross-sectional studies was used to assess the quality of the studies in terms of its inclusion. The total score for the modified Newcastle–Ottawa scale for cross-sectional studies used was nine (9) stars as a maximum for the overall scale with the minimum of zero, and a study was considered to be a high quality if 7 was achieved from 9 and medium if 5 was achieved from 9 [46].

## Operational definitions

**Compliance.**    Pregnant woman who took ≥65% of the total prescribed IFA supplementation per month was considered as a good compliance; whereas the opposite of this rate is true for non-compliance types [1].

**Trimester.**    Was defined as the number of weeks during pregnancy period. That is {(1st, 1–12 weeks), (2nd, 13–26 weeks), and (3rd, 27–40 weeks) [20].

**Outcome interests.**    The primary outcome of this study was a compliance of IFAS during the pregnancy period of the woman. In this study, Potential factors affecting the compliance of IFA supplementation includes: counseling on IFAS, knowledge of IFA supplementation, knowledge of anemia, fourth visit for ANC and early registration for ANC. Thus, Knowledge on anemia was defined as those who heard and knew at least one of the signs and symptoms of this public health problem, anemia. However, information about IFAS was measured by asking questions of knowledge related to IFAS (Iron/folate drug, health benefit of IFAS for the fetus and child to identify whether they believe the risky of taking IFAS and know for how long they should take the IFAS or not. Another key related issue is early registration to ANC clinic which was measured based on number of pregnant women who visit the ANC clinic before 16 weeks of gestation. Also, counseling on IFAS was defined as Women who have received information on the IFA supplementation. The final clarification is the fourth visit for ANC which is defined as pregnant women who received antenatal care (ANC) 4 or more times during the pregnancy period.

## Statistically analysis

The extracted data was copied to Microsoft excel to be exported to review manager version 5.3 and the compressive meta-analysis version 2 software for careful analysis. Accordingly, statistical description related to IFAS and its determinants were performed. To evaluate the existence of statistical heterogeneity, the Publications bias was tested by funnel plot and empirically through Egger's regression test. Moreover, the degree of trustworthiness was contemplated. Consequently, the heterogeneity of studies was computed using the I-squared statistic. In this process, 25% was signified as low, 50% moderate and 75% as high heterogeneity score. Also subgroup analysis was executed by the study sub-region and study type (Community based

and / or facility based). Therefore, the effect of particular predictor's variables which consist of: Counseling on IFAS, knowledge of IFA supplementation, knowledge of anemia, fourth visit for ANC and early registration for ANC were estimated and the result of the Meta-analysis revealed forest pilot and Odd Ratio (OR) with 95% of CI.

## Result

### Studies selection

Based on the objectives set for this work, we identified different studies related to the prevalence and determinants of IFA supplementation for the inclusion in meta-analysis before directly move to the other detail part of this paper. Accordingly, we found 1156 completed studies published on international journals. From these, 1113 of them were excluded for they were not satisfying the criterion of inclusion set in this study. However, 43 articles were scrutinized and carefully chosen from those studies for their eligibility. Out of these 20 studies were rejected due to their poor statistical reports and defect of data observed in each of them. Finally, only 20 studies were added in this analysis for their neatness and clear justification (**Fig 1**).

### Characteristics of included studies

Twenty-three Cross-Sectional studies from different countries of Africa were included in the meta-analysis [23–45] (**Table 1**). Out of those four of them (17.4%) were from Kenya, two (8.7%) from Malawi, two (8.7%) from Nigeria, one (4.7%) from Mozambique, one (4.3%) from Senegal, one (4.3%) from South Africa, eleven (47.8%) from Ethiopia and one (4.3%) from Uganda. Among those the highest sample size was observed in studies conducted in Malawi [23] which was equal 10750 and the lowest was found in South Africa, 57 [39]. The mean age of the respondents was 27 years. Out of twenty-three studies incorporated in this review and analysis, though eighteen studies [25, 27–33, 35–38, 40–45] were conducted on facility based, five of them [23, 24, 26, 34, 39] were done on community based work. However, all studies considered definition of good adherence toward IFA as a pregnant woman who received $\geq$65% of the total recommended IFA supplementation per month, but the reverse of this is true for non-adherence types [1]. Lastly, (**Table 2**) displays assessment of quality of studies included in systematic review and Meta-analysis.

### Compliance with Iron and Folic-Acid Supplementation

From the analysis made, we can understand that the lowest prevalence of compliance (10.6%) was observed in Kenya [38], but the highest compliance, which is equal to (79%) was observed in study conducted in Mozambique [27]. On the other hand, the pooled compliance with Iron and Folic-Acid Supplementation (IFAS) amongst pregnant women in SSA was 39.2% (95% CI = 0.308–0.483) (**Fig 2**). Moreover, the heterogeneity test displayed is equal to $I^2$ = 99.27% and the statistical suggestion for heterogeneity is P<0.000). Hence, the random-effect analysis was the secondhand of the analysis. Therefore, the Bag's and Egger's test for publication bias indicated that there was no statistical suggestion for publication bias. That is p-value = 0.428 and 0.575 respectively (**See S1 Fig**).

### Subgroup analysis

A subgroup analysis was done by classifying studies based on corresponding sub-regional location in Sub-Saharan Africa in order to compute and relate the prevalence of compliance with IFA supplementation focusing on athwart various participants' characteristics. Based on this, the lowest prevalence of compliance with IFA supplementation in pregnancy was documented in Eastern Africa, (34.2%) (CI: 0.264, 0.430) and the highest prevalence of compliance with

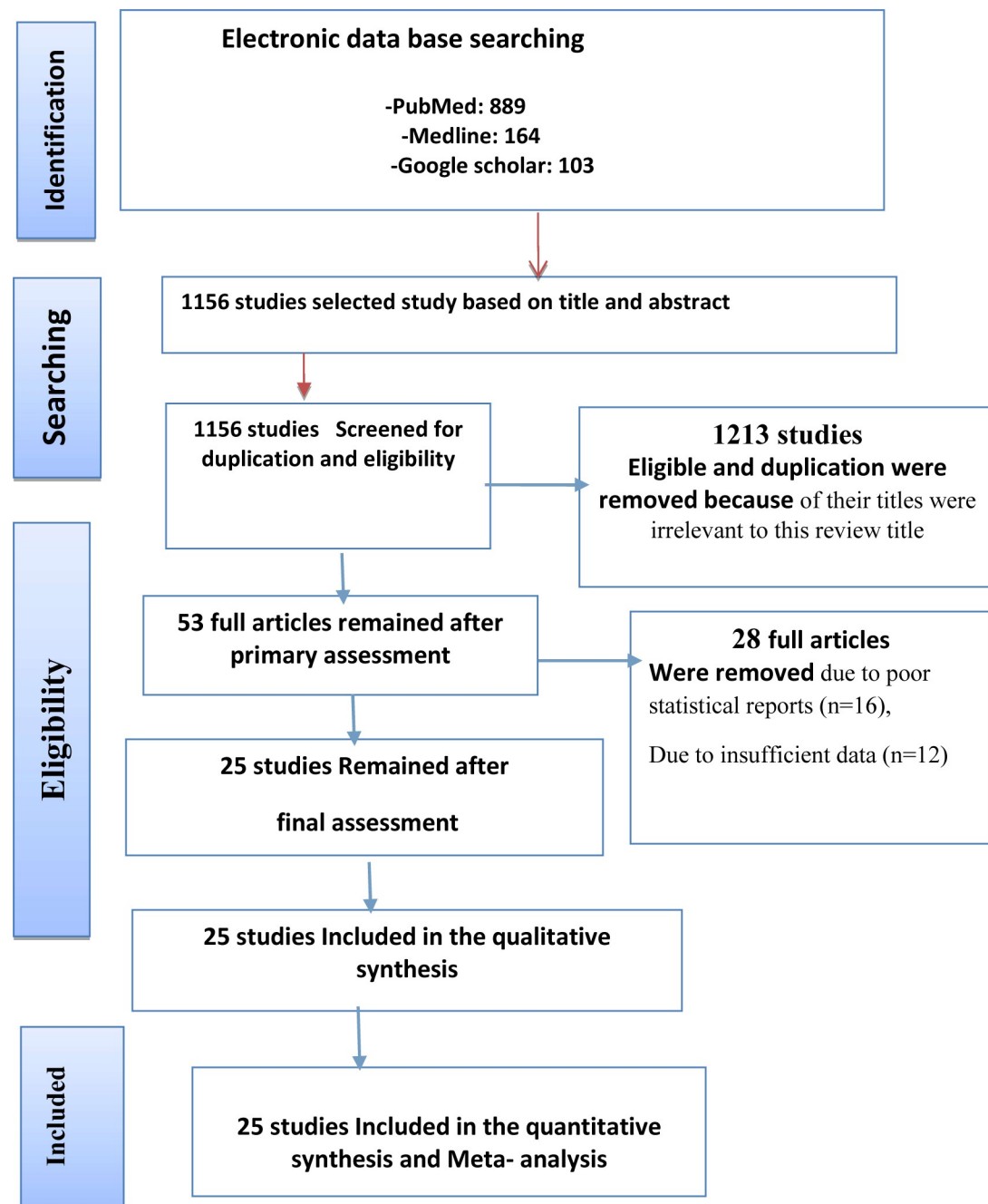

**Fig 1. Flow diagram of the studies included in the meta-analysis.**

IFA supplementation was recognized in Western Sub-Saharan, (49.3% (CI: 0.203, 0.788). However, a greater prevalence of compliance, which is equal to (44.3%%) was detected in studies conducted at facility level than community level (CI: 0.337, 0.554) (**Table 3**).

## Counseling on Iron and Folic Acid Supplementation

Out of twenty-three chosen studies conducted on the area of the key concern and included in the meta-analysis, in nine of them [25, 28, 30, 31, 34, 35, 37, 38, 45] it was documented that

**Table 1. Characteristics of studies included in systematic review of prevalence compliance with IFAS among pregnant women in Sub-Saharan Africa.**

| Author | Region | Study design | Study type | Sample Size | Prevalence in % | Compliance IFAS | No compliance IFAS |
|---|---|---|---|---|---|---|---|
| Abinet Arega Sadore et al, 2015 | Ethiopia | Cross sectional | Community based | 296 | 39.2 | 116 | 180 |
| Titilayo A. et al, 2016 | Malawi | Cross sectional | Community based | 10750 | 37.2 | 3999 | 6751 |
| Chikakuda A. et al, 2018 | Malawi | Cross sectional | Facility based | 213 | 18.3 | 39 | 174 |
| Bekele Taye et al, 2015 | Ethiopia | Cross sectional | Community based | 628 | 20.4 | 128 | 500 |
| BI Nwaru et al, 2014 | Mozambique | Cross sectional | Facility based | 4326 | 79 | 3418 | 908 |
| Demis et al, 2019 | Ethiopia | Cross sectional | Facility based | 422 | 43.1 | 182 | 240 |
| Agegnehu G. et al, 2018 | Ethiopia | Cross sectional | Facility based | 418 | 28.7 | 120 | 298 |
| Gebremariam et al, 2019 | Ethiopia | Cross sectional | Facility based | 241 | 40 | 96 | 145 |
| Getachew et al, 2018 | Ethiopia | Cross sectional | Facility based | 320 | 64.7 | 207 | 113 |
| Dessie G. et al, 2018 | Ethiopia | Cross sectional | Facility based | 348 | 19 | 66 | 282 |
| Juma M et al, 2015 | Kenya | Cross sectional | Facility based | 352 | 18.3 | 64 | 288 |
| K. Niang et al, 2017 | Senegal | Cross sectional | Community based | 1442 | 51 | 735 | 707 |
| Kamau et al, 2018 | Kenya | Cross sectional | Facility based | 364 | 33.7 | 123 | 241 |
| Kiwanuka et al, 2017 | Uganda | Cross sectional | Facility based | 370 | 11.6 | 43 | 327 |
| Lucy Nyandia Gathigi, 2011 | Kenya | Cross sectional | Facility based | 264 | 10.6 | 28 | 236 |
| LYNETTE AOKO DINGA, 2013 | Kenya | Cross sectional | Facility based | 200 | 24.5 | 49 | 151 |
| Mbhenyane et al, 2017 | South Africa | Cross sectional | Community based | 57 | 90 | 51 | 6 |
| Molla et al, 2019 | Ethiopia | Cross sectional | Facility based | 348 | 52.9 | 184 | 164 |
| Niguse and Murugan, 2018 | Ethiopia | Cross sectional | Facility based | 296 | 59.8 | 177 | 119 |
| Onyeneho et al, 2016 | Nigeria | Cross sectional | Facility based | 1500 | 33 | 495 | 1005 |
| Shewasinad S, et al, 2017 | Ethiopia | Cross sectional | Facility based | 326 | 70.6 | 230 | 96 |
| Tarekegn et al, 2019 | Ethiopia | Cross sectional | Facility based | 395 | 28.0 | 111 | 284 |
| Ugwu, et al, 2012 | Nigeria | Cross sectional | Facility based | 396 | 65.9 | 261 | 135 |

counseling on IFA supplementation was associated with compliance to IFA supplementation in pregnancy. Moreover, the result from meta-analysis also revealed that pregnant women who had received counseling on IFAS during pregnancy were 1.96 times more likely than those who had adherence to IFAS as compared to those who had not received counseling, [OR: 1.96, 95% CI (1.76-, 5.93)]. Thus, the heterogeneity test revealed $I^2$ = 92% and the statistical evidence of this is P<0.00001). From this we can understand that the random-effect analysis was the secondhand one. Thus, the Bag's and Egger's test for publication bias indicated that there is no statistical evidence of Publication bias. That is their p-value = 0.284 and = 0.754 respectively (**Fig 3**).

## Knowledge of Iron Folic Acid Supplementation (IFAS)

The association of lack of knowledge on Iron Folic-Acid Supplementation and risk of developing noncompliance to IFAS during pregnancy was stated in nine chosen studies [25, 26, 28, 30, 32, 35, 37, 38, 40]. Thus, the result of meta-analysis from those exhibited that women who had knowledge of IFA supplementation were 2.71 times more likely to have compliance to IFA Supplementation compared to those who had no knowledge of IFA Supplementation [OR: 2.71, 95% CI (1.33, 5.54)]. Therefore, the heterogeneity test indicated (I2 = 94%) and statistical evidence of this heterogeneity was P<0.0000). Hence, the random- analysis was the secondhand one again. Finally, the Bag's and Egger's test for publication bias indicated the absence of statistical evidence of Publication bias. That is their p-values are equal to 0.465 and 0.786 respectively (**Fig 4**).

**Table 2. Assessment of quality of studies included in systematic review of prevalence compliance with IFAS among pregnant women in Sub-Saharan Africa.**

| Studies | Representativeness of the sample | | | Sample size | | | Non-respondents | Ascertainment of the exposure (risk factor) | | | Confounding factors are controlled | | Ass. of the outcome | | Statistical test | | | Total |
|---|---|---|---|---|---|---|---|---|---|---|---|---|---|---|---|---|---|---|
| | All subjects or random sampling* | Non random sampling* | No description of sampling strategy | Justified and satisfactory* | Not Justified | Satisfactory* | Unsatisfactory | Validated measurement tool** | non Validated measure or tool is available* | No description of Validated measurement tool | Study controls important factor* | Study controls Additional factor* | Independent blind assessment** | Record linkage** | Self-report* | No description | statistical test used to analyze the data is clearly described* | |
| Abinet Arega Sadore et al, 2015 | * | | | * | | * | | * | | | | | * | | | | | 6 |
| Titilayo A.et al, 2016 | * | | | * | | * | | * | | | * | | ** | | | | | 7 |
| Chikakuda A. et al, 2018 | * | | | * | | * | | ** | | | * | | ** | | | | | 8 |
| Bekele Taye et al, 2015 | * | | | * | | * | | ** | | | | | * | | | | * | 7 |
| BI Nwaru et al, 2014 | * | | | * | | * | | ** | | | * | | * | | | | * | 7 |
| Demis et al, 2019 | * | | | * | | * | | ** | | | * | | ** | | | | * | 9 |
| Agegnehu G. et al, 2018 | * | | | * | | * | | ** | | | * | | ** | | | | * | 8 |
| Gebremariam et al, 2019 | * | | | * | | * | | * | | | * | | ** | | | | * | 8 |
| Getachew et al, 2018 | * | | | * | | * | | * | | | * | | * | | | | * | 7 |
| Dessie G. et al, 2018 | * | | | * | | * | | * | | | * | | ** | | | | * | 8 |
| Juma M et al, 2015 | * | | | * | | * | | * | | | * | | ** | | | | * | 7 |
| K. Niang et al, 2017 | * | | | * | | * | | ** | | | * | | ** | | | | * | 9 |
| Kamau et al, 2018 | * | | | * | | * | | ** | | | * | | ** | | | | * | 9 |
| Kiwanuka et al, 2017 | * | | | * | | * | | ** | | | * | | ** | | | | | 8 |
| Lucy Nyandia Gathigi, 2011 | * | | | * | | * | | ** | | | * | | ** | | | | * | 9 |
| LYNETTE AOKO DINGA, 2013 | * | | | * | | * | | ** | | | * | | * | | | | * | 7 |
| Mbhenyane et al, 2017 | * | | | * | | * | | ** | | | * | | ** | | | | | 8 |
| Molla et al, 2019 | * | | | * | | * | | ** | | | * | | * | | | | * | 8 |
| Niguse and Murugan, 2018 | * | | | * | | * | | ** | | | * | | ** | | | | * | 9 |
| Onyeneho et al, 2016 | * | | | * | | * | | * | | | * | | ** | | | | * | 6 |
| Shewasinad S, et al, 2017 | * | | | * | | * | | ** | | | * | | ** | | | | * | 9 |
| Tarekegn et al, 2019 | * | | | * | | * | | ** | | | * | | ** | | | | * | 8 |
| Ugwu, et al, 2012 | * | | | * | | * | | ** | | | * | | ** | | | | * | 9 |

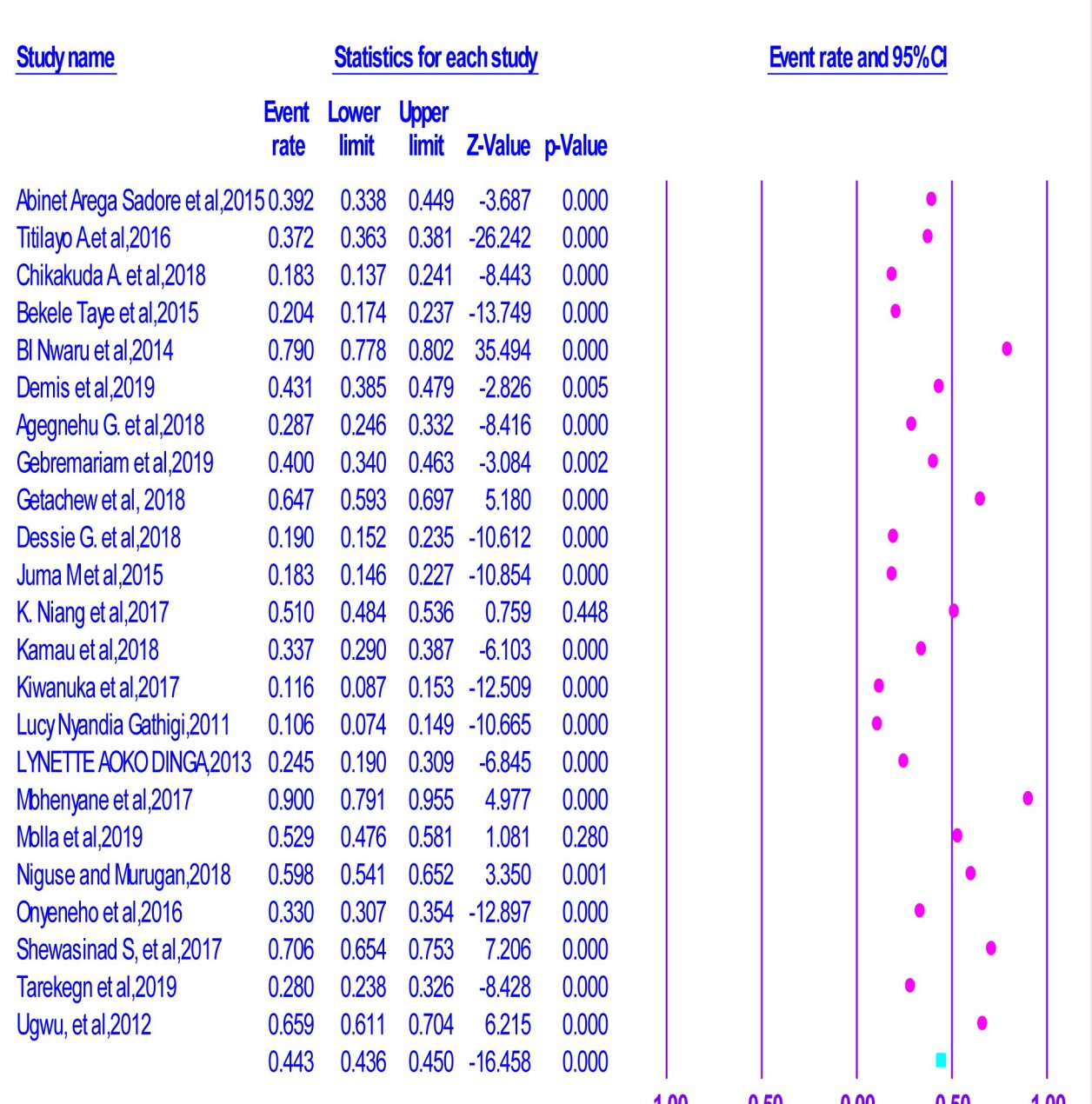

| Study name | Statistics for each study | | | | | Event rate and 95% CI |
|---|---|---|---|---|---|---|
| | Event rate | Lower limit | Upper limit | Z-Value | p-Value | |
| Abinet Arega Sadore et al,2015 | 0.392 | 0.338 | 0.449 | -3.687 | 0.000 | |
| Titilayo A.et al,2016 | 0.372 | 0.363 | 0.381 | -26.242 | 0.000 | |
| Chikakuda A. et al,2018 | 0.183 | 0.137 | 0.241 | -8.443 | 0.000 | |
| Bekele Taye et al,2015 | 0.204 | 0.174 | 0.237 | -13.749 | 0.000 | |
| BI Nwaru et al,2014 | 0.790 | 0.778 | 0.802 | 35.494 | 0.000 | |
| Demis et al,2019 | 0.431 | 0.385 | 0.479 | -2.826 | 0.005 | |
| Agegnehu G. et al,2018 | 0.287 | 0.246 | 0.332 | -8.416 | 0.000 | |
| Gebremariam et al,2019 | 0.400 | 0.340 | 0.463 | -3.084 | 0.002 | |
| Getachew et al, 2018 | 0.647 | 0.593 | 0.697 | 5.180 | 0.000 | |
| Dessie G. et al,2018 | 0.190 | 0.152 | 0.235 | -10.612 | 0.000 | |
| Juma M et al,2015 | 0.183 | 0.146 | 0.227 | -10.854 | 0.000 | |
| K. Niang et al,2017 | 0.510 | 0.484 | 0.536 | 0.759 | 0.448 | |
| Kamau et al,2018 | 0.337 | 0.290 | 0.387 | -6.103 | 0.000 | |
| Kiwanuka et al,2017 | 0.116 | 0.087 | 0.153 | -12.509 | 0.000 | |
| Lucy Nyandia Gathigi,2011 | 0.106 | 0.074 | 0.149 | -10.665 | 0.000 | |
| LYNETTE AOKO DINGA,2013 | 0.245 | 0.190 | 0.309 | -6.845 | 0.000 | |
| Mbhenyane et al,2017 | 0.900 | 0.791 | 0.955 | 4.977 | 0.000 | |
| Molla et al,2019 | 0.529 | 0.476 | 0.581 | 1.081 | 0.280 | |
| Niguse and Murugan,2018 | 0.598 | 0.541 | 0.652 | 3.350 | 0.001 | |
| Onyeneho et al,2016 | 0.330 | 0.307 | 0.354 | -12.897 | 0.000 | |
| Shewasinad S, et al,2017 | 0.706 | 0.654 | 0.753 | 7.206 | 0.000 | |
| Tarekegn et al,2019 | 0.280 | 0.238 | 0.326 | -8.428 | 0.000 | |
| Ugwu, et al,2012 | 0.659 | 0.611 | 0.704 | 6.215 | 0.000 | |
| | 0.443 | 0.436 | 0.450 | -16.458 | 0.000 | |

**Fig 2. Forest plot displaying association of counselling on Iron and Folic Acid Supplementation with compliance with IFAS among pregnant women in Sub-Saharan Africa.**

## Knowledge of anemia during pregnancy

The important analysis was focus on the association of lack of knowledge on Iron Folic-Acid Supplementation and risk of developing noncompliance to IFAS during pregnancy. This was stated in seven different studies [25, 26, 31, 37, 38, 40, 45]. The result of meta-analysis from those paper showed that women who had knowledge about anemia were 5.42 times more likely to have compliance to IFAS in their course of pregnancy as compared to those who had

**Table 3. Subgroup analysis of prevalence compliance with IFAS among pregnant women in Sub-Saharan Africa.**

| Subgroup | No. of included studies | Prevalence(95% CI) | Heterogeneity Statistics | Tau Squared | P value | I² |
|---|---|---|---|---|---|---|
| By Sub- region | | | | | | |
| Eastern Africa | 17 | 34.2(0.264,0.430) | 787.551 | 0.594 | <0.000 | 97.968 |
| Southern Africa | 4 | 58.3(0.277,0.836) | 1995.397 | 1.693 | <0.000 | 99.850 |
| Western Africa | 2 | 49.3(0.203,0.788) | 131.117 | 6.927 | <0.000 | 99.927 |
| Overall | | | | | <0.000 | |
| By study type | | | | | | |
| Facility based | 18 | 37(0.256,.0501) | 2333.651 | 0.696 | <0.000 | 99.272 |
| Community based | 5 | 44.3(0.337,0.554) | 218.345 | 0.230 | <0.000 | 98.16 |

no knowledge of anemia, [OR5.42, 95% CI (1.52, 19.43)]. The heterogeneity test indicated that ($I^2$ is equal to 97% and the statistical evidence of this is P<0.0000). Hence, the random-effect analysis was secondhand one, but the Bag's and Egger's test for publication bias indicated that there is no statistical evidence of Publication bias, which is equal to p-value = 0.283 and 0.682 respectively (Fig 5).

## Fourth visit for Antenatal Care (ANC)

The association between lack of fourth visit for antenatal care and risk of developing noncompliance to IFAS during pregnancy was stated in eight studies [25, 28, 30, 31, 36, 37, 40, 43]. Accordingly, the result from meta-analysis from those revealed that women who had fourth visit for ANC in pregnancy were 1.54 times more likely to have compliance to IFAS during

**Fig 3. Forest plot displaying association of counseling on Iron and Folic Acid Supplementation with compliance with IFAS among pregnant women in Sub-Saharan Africa.**

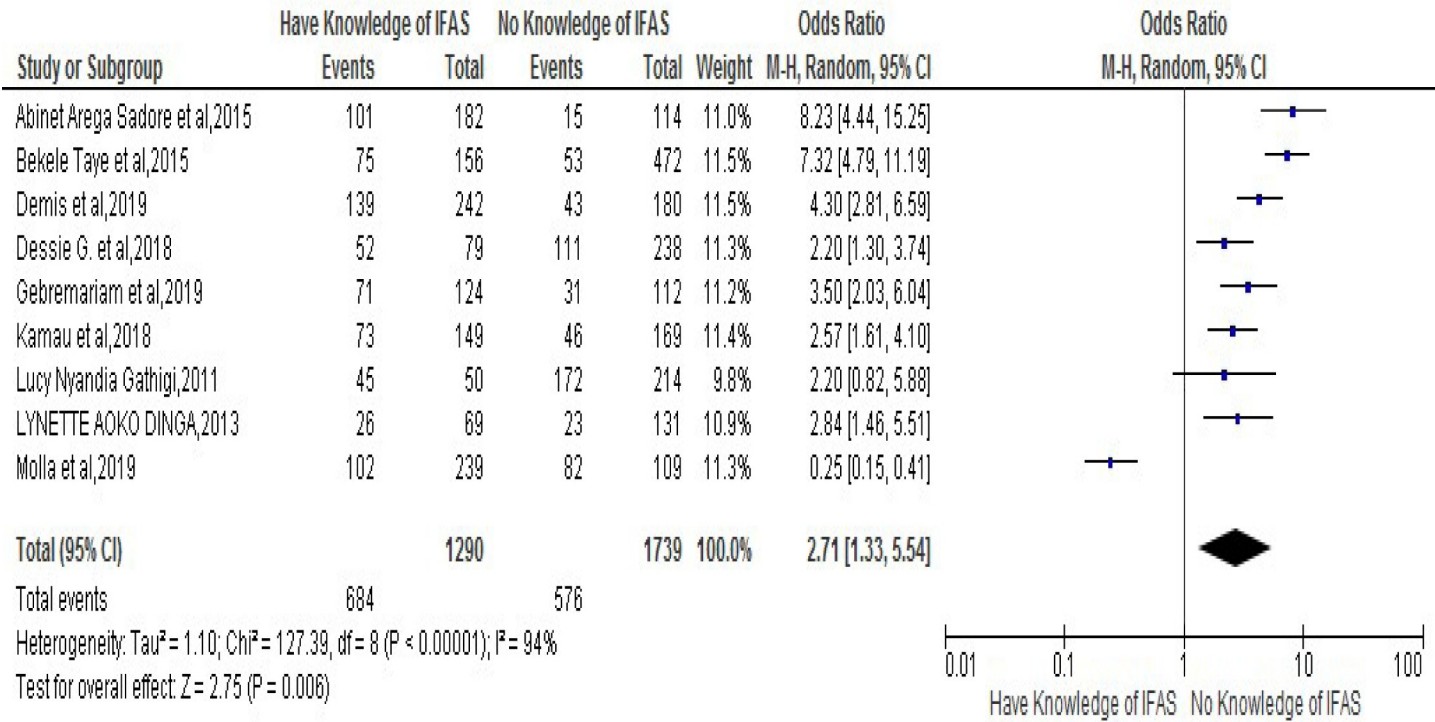

**Fig 4. Forest plot displaying association of knowledge of Iron Folic-Acid Supplementation (IFAS) with compliance with IFAS among pregnant women in Sub-Saharan Africa.**

**Fig 5. Forest plot displaying association of knowledge of Anaemia during pregnancy with compliance with IFAS among pregnant women in Sub-Saharan Africa.**

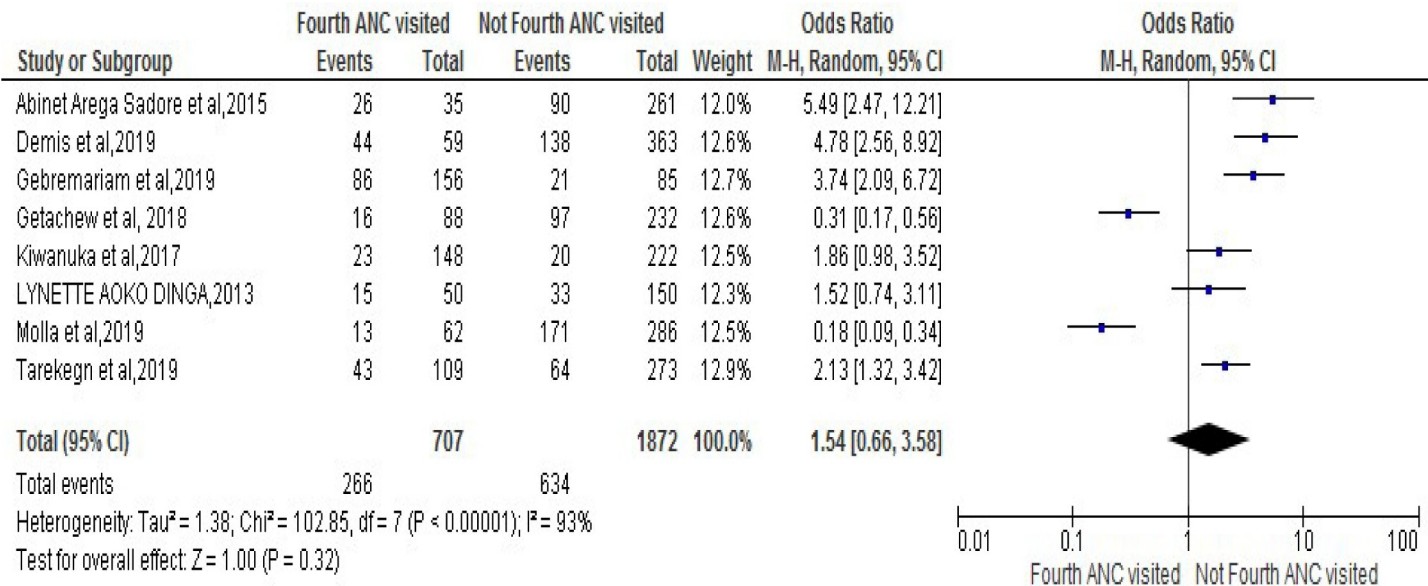

**Fig 6. Forest plot displaying association of Fourth visit for antenatal care (ANC) with compliance with IFAS among pregnant women in Sub-Saharan Africa.**

pregnancy as compared to those had no fourth visit for ANC [OR 1.54, 95% CI (0.66,3.58.43)]. However, the heterogeneity test indicated $I^2$ = 93% and statistical evidence of this was <0.00001). Thus, random-effect analysis was the secondhand one. On the other hand, the Bag's and Egger's test for publication bias indicated that there was no statistical evidence of Publication bias. That is their p-value is equal to 0.980 and 0.982 consecutively (**Fig 6**).

## Discussion

Anemia is one of the globally top causes of frailty, the highest universal problem and identified public health matters in Sub-Saharan Africa [1]. Evidences from the meta- analysis stated above suggested that almost 38% (32 million) women were victims of anemia in their course of pregnancy in the word.

   The systematic review and meta-analysis presented in this paper showed the magnitude of compliance to IFAS in sub-Saharan-Africa and its determinants. Accordingly, the key finding of analysis exhibited the adherence to IFAS in pregnancy prevalence in Sub-Saharan Africa in which the pooled prevalence was equal to 39.2%; however, the heterogeneity test indicates its statistical evidence which was elucidated by difference in geographic location, for instance, Eastern Africa, Southern Africa and western Africa types of study which was focused on community and facility based type. In relation to this analysis, a study conducted in Iran stated that the prevalence of compliance with IFAS among pregnant women was 71.6% % [47]. Similarly, in Egypt more than one-third of the pregnant women were not taking iron-folate tablets during their pregnancy [48]. Moreover, another findings in India revealed that compliance with IFAS among pregnant women was 64% [49]. Therefore, the combined prevalence of compliance with IFAS among pregnant women in this study was lower than those studies conducted in Iran and India which is comparable with study done in Egypt. Thus, our finding varies significantly from studies conducted in advanced countries. This pointed out that women of Sub-Saharan Africa have low compliance to IFAS in the course of their pregnancy.

   The present pooled meta-analysis revealed that pregnant women who had knowledge of IFAS were almost three times more likely to have compliance with IFAS during pregnancy

when compared to those women who had no knowledge of IFAS. Also pregnant women who had knowledge of anemia during pregnancy were five times more likely to have compliance of IFAS as compared to those women who had no knowledge of Anemia. In relation to this, an investigation achieved in Egypt showed that there was a high statistically significant which has a positive correlation between women's score of knowledge about folic acid, iron and anemia and their score of compliance to iron /folate supplementation during pregnancy [48].

On the other hand, the pooled effect of present meta-analysis presented that pregnant women who had fourth visit for ANC during pregnancy were almost two times more likely to have compliance with IFAS during their pregnancy as compared to those women who had no fourth visit for ANC. Similarly, an investigation on this issue in five Asian counties like, India, Indonesia, Nepal, Pakistan, and the Philippines showed that the pregnant women who received at least three antenatal care visits were much more likely to adhere at least 90 days of iron tablet or syrup or iron and folic acid tablets supplementation. Moreover, it also suggested that antenatal care-seeking visits seem to be a particularly effective ways of reaching women in increasing the likelihood of uptake of iron only or iron and folic acid supplements [49].

However, the study lacks representativeness since there was no data found from some Sub-Saharan African counties. Also there were no adequate studies incorporated in the analysis. Thus, this shortcoming could trouble the over-all prevalence of compliance to IFAS among pregnant women in Sub-Saharan Africa.

## Strength and limitation

In this review, an extensive exploration method and more than one reviewer had taken part in all courses of review process. To do so, PRISMA guideline was carefully tracked throughout the review procedure. However, the analysis has its own defects because of a number of factors. For instance, firstly compliance to IFA supplementation was defined in various studies in different way. Secondly, study lacks representativeness since no data was found from some of Sub-Saharan African counties. Thirdly, some studies have been omitted due to their poor statistical reports, their small sample size, their inadequate data and their qualitative nature of the studies. Finally, since only the cross sectional studies were involved in the analysis, the outcome variable may possibly be affected by confounding variable. Therefore, those limitations might affect the overall prevalence of compliance to IFA supplementation in pregnancy in Sub-Saharan Africa.

## Conclusion: Implication for practice and future research

Our findings suggest the prevalence of compliance to IFAS among pregnant women in Sub-Saharan Africa (SSA) low. Thus, millions of women in SSA are still lack access to life saving IFA supplementations during pregnancy and they are at risk. Predictor comprises: knowledge about anemia, knowledge about IFAS, counseling on IFAS and receiving fourth antenatal care visit were positively correlated with compliance to IFAS statistically. This demands the careful appraisal effect of prevention work for functioning policy, programs and plan nutrition intrusions for refining maternal dietary intake during gestation period. Moreover, dietary education intrusions should be planned to meet the needs of pregnant women to improve their dietary practice.

Consequently, health care providers ought to provide dietary guidance to enhance antenatal care service frequently. For example, the suggestion from WHO reference showed that good supplementation requires building guarantee that a pregnant women is well prescribed for IFAS (lowest 90 tablets) and timely booking her prior to 12 weeks of gestation period. Equally important, the management ought to assign dietarian at all level of health system. However,

exceptional training should be prearranged for health care providers who are at frontline antenatal care service at each level of health system in order to improve adherence of IFA supplementation at work place. Accordingly, the result of this study might be essential for policy makers of Africa, exclusively for maternal and child health care. Finally, further studies should be conducted in the area to search and produce extra suggestions to alleviate the problem.

## Supporting information

**S1 Fig. Funnel plot displaying publication bias of prevalence of compliance with IFAS among pregnant women in Sub-Saharan Africa.** Description of figure: This figure presents, Bag's and Egger's test for publication bias showed no statistical evidence of publication bias. (DOCX)

**S1 File.**
(DOC)

**S2 File.**
(DOCX)

**S3 File.**
(DOCX)

## Author Contributions

**Conceptualization:** Meseret Belete Fite, Abera Kenay Tura, Tesfaye Assebe Yadeta.

**Data curation:** Meseret Belete Fite, Abera Kenay Tura, Tesfaye Assebe Yadeta.

**Formal analysis:** Meseret Belete Fite, Abera Kenay Tura.

**Methodology:** Meseret Belete Fite, Abera Kenay Tura, Tesfaye Assebe Yadeta.

**Software:** Meseret Belete Fite, Lemessa Oljira, Abera Kenay Tura.

**Supervision:** Meseret Belete Fite, Kedir Teji Roba, Lemessa Oljira, Abera Kenay Tura, Tesfaye Assebe Yadeta.

**Validation:** Meseret Belete Fite, Kedir Teji Roba, Lemessa Oljira, Abera Kenay Tura, Tesfaye Assebe Yadeta.

**Visualization:** Meseret Belete Fite, Lemessa Oljira, Tesfaye Assebe Yadeta.

**Writing – original draft:** Meseret Belete Fite, Kedir Teji Roba, Lemessa Oljira, Abera Kenay Tura, Tesfaye Assebe Yadeta.

**Writing – review & editing:** Meseret Belete Fite, Kedir Teji Roba, Lemessa Oljira, Abera Kenay Tura, Tesfaye Assebe Yadeta.

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
