## [Decision Letter · Decision Letter 0]

7 Oct 2020

PONE-D-20-25562

Compliance with Iron and folic acid supplementation (IFAS) and associated factors among pregnant women Sub-Saharan Africa: A systematic review and meta-analysis

PLOS ONE

Dear Dr. Fite,

Thank you for submitting your manuscript to PLOS ONE. After careful consideration, we feel that it has merit but does not fully meet PLOS ONE’s publication criteria as it currently stands. Therefore, we invite you to submit a revised version of the manuscript that addresses the points raised during the review process.

Two experts in the field handled your manuscript. ALL of the reviewers' comments need to be addressed in your revised manuscript.

We look forward to receiving your revised manuscript.

Kind regards,

Frank T. Spradley

Academic Editor

PLOS ONE

2. Please confirm that you have included all items recommended in the PRISMA checklist including the full electronic search strategy used to identify studies with all search terms and limits for at least one database.

3. Please ensure that you refer to Figure 6 in your text as, if accepted, production will need this reference to link the reader to the figure.

4. Please upload a copy of Figure 4, to which you refer in your text on page 5. If the figure is no longer to be included as part of the submission please remove all reference to it within the text. (Fig 5 2x)

5.Thank you for submitting the above manuscript to PLOS ONE. During our internal evaluation of the manuscript, we found significant text overlap between your submission and the following previously published works.

- https://bmchematol.biomedcentral.com/track/pdf/10.1186/s12878-018-0124-1

- https://archpublichealth.biomedcentral.com/articles/10.1186/s13690-019-0356-y?optIn=false

Please revise the manuscript to rephrase the duplicated text, cite your sources, and provide details as to how the current manuscript advances on previous work. Please note that further consideration is dependent on the submission of a manuscript that addresses these concerns about the overlap in text with published work.

Reviewers' comments:

Reviewer's Responses to Questions

**Comments to the Author**

1. Is the manuscript technically sound, and do the data support the conclusions?

Reviewer #1: Partly

Reviewer #2: Yes

2. Has the statistical analysis been performed appropriately and rigorously? 

Reviewer #1: Yes

Reviewer #2: Yes

3. Have the authors made all data underlying the findings in their manuscript fully available?

Reviewer #1: Yes

Reviewer #2: Yes

4. Is the manuscript presented in an intelligible fashion and written in standard English?

Reviewer #1: Yes

Reviewer #2: No

5. Review Comments to the Author

Reviewer #1: It was interesting study where authors summarized the compliance with Iron and folic acid supplementation (IFAS) and potential associated factors among pregnant women in Africa. Authors used meta-analysis to summarize this compliance, so results were valuable to some extent. This summary may be important for policy makers of Africa especially for maternal and child health care. However, some technical issues should be further addressed, which could improve the manuscript.

1. Due to limited studies included, the conclusion from this study was cautious. And studies included in analysis from one African country were limited (most of studies were form Ethiopia) and might be not a representative analysis for this country. Authors should deeply analyze this issue in the discussion.

2. I think this was a meta-analysis, not a complete systematic review because lots of studies from different designs were excluded. Authors just made a mathematic analysis based on limited data. So I suggest that “A systematic review” in the title might be deleted.

3. In the section of methods, authors said in line 121-222 that “Secondary outcome include: Counselling on IFAS, knowledge of IFA supplementation, knowledge of anaemia, fourth visit for ANC and early registration for ANC”. I disagree with it. Actually, authors still studied on primary outcome----a compliance of IFAS during pregnancy. This was further analysis about potential factors affecting the compliance of IFAS during pregnancy. Therefore, this sentence should be revised.

4. Authors should report the definition of Counselling on IFAS, knowledge of IFA supplementation, knowledge of anaemia, fourth visit for ANC and early registration for ANC in the section of methods. Different studies could have different definitions.

5. Authors should present the results on characteristics and assessment of quality of study in main text not supplementary file because they are one of main results for meta-analysis.

6. Due to too small sample size (study 39), maybe sensitive analysis is suggested when excluding this study.

7. Minor comments: for figure 2, the name of pooled rate should be added in the right place.

Reviewer #2: 1. The background section of the abstract is a copy/paste of some introductory parts in the main manuscript. This is not a good practice. it is also very long and needs to shortened.

2. Does anemia cause disability? This needs to be clarified and reference given.

3. The definition of compliance is completely missing. This need to be included at appropriate section or subsection. Was the definition uniform across all the publication included for meta-analysis?

4. Why was the DHS data not used. DHS data collection is uniform across countries and would have been the most suitable. The authors need to explain why DHS data was not used or was excluded from this analysis.

5. The conclusion both in the abstract and manuscript does not highlight on the future research opportunities and policy considerations. Only programmatic implications are highlighted.

6. The paper will benefit from review for grammar, typos and clarity.

6. PLOS authors have the option to publish the peer review history of their article (what does this mean?). If published, this will include your full peer review and any attached files.

Reviewer #1: No

Reviewer #2: No

---

## [Decision Letter · Decision Letter 1]

29 Dec 2020

PONE-D-20-25562R1

Compliance with Iron and folic acid supplementation (IFAS) and associated factors among pregnant women Sub-Saharan Africa: A systematic review and meta-analysis

PLOS ONE

Dear Dr. Fite,

Thank you for submitting your manuscript to PLOS ONE. After careful consideration, we feel that it has merit but does not fully meet PLOS ONE’s publication criteria as it currently stands. Therefore, we invite you to submit a revised version of the manuscript that addresses the points raised during the review process.

There are remaining comments that need to be addressed. Notably, grammatical and spelling errors, if not corrected, are enough to prohibit acceptance on this article. Please address ALL of the reviewers' comments in your revised manuscript.

We look forward to receiving your revised manuscript.

Kind regards,

Frank T. Spradley

Academic Editor

PLOS ONE

Reviewers' comments:

Reviewer's Responses to Questions

**Comments to the Author**

1. If the authors have adequately addressed your comments raised in a previous round of review and you feel that this manuscript is now acceptable for publication, you may indicate that here to bypass the “Comments to the Author” section, enter your conflict of interest statement in the “Confidential to Editor” section, and submit your "Accept" recommendation.

Reviewer #1: All comments have been addressed

Reviewer #2: (No Response)

2. Is the manuscript technically sound, and do the data support the conclusions?

Reviewer #1: Yes

Reviewer #2: Yes

3. Has the statistical analysis been performed appropriately and rigorously? 

Reviewer #1: Yes

Reviewer #2: Yes

4. Have the authors made all data underlying the findings in their manuscript fully available?

Reviewer #1: Yes

Reviewer #2: Yes

5. Is the manuscript presented in an intelligible fashion and written in standard English?

Reviewer #1: Yes

Reviewer #2: No

6. Review Comments to the Author

Reviewer #1: Authors have addressed most of my concerns, and the manuscript has been improved. I have not addtional comments.

Reviewer #2: It would be good to explain in the resubmission how each comment has been addressed. For instance:

This comment has not been addressed: 'The background section of the abstract is a copy/paste of some introductory parts in the main manuscript. This is not a good practice. it is also very long and needs to shortened' the opening statement of the background still reads the same as the statement in the introduction

The definition of compliance is completely missing. This need to be included at appropriate section or subsection. Was the definition uniform across all the publication included for meta-analysis? WHERE IS THE OPERATIONAL DEFINITION OF COMPLIANCE?

The paper will STILL benefit from review for grammar, typos and clarity. THE TYPOS ARE STILL VISIBLE EVEN IN THE ABSTRACT

7. PLOS authors have the option to publish the peer review history of their article (what does this mean?). If published, this will include your full peer review and any attached files.

Reviewer #1: No

Reviewer #2: No

---

## [Editor Report · Decision Letter 2]

25 Mar 2021

Compliance with Iron and folic acid supplementation (IFAS) and associated factors among pregnant women Sub-Saharan Africa:  A systematic review and meta-analysis

PONE-D-20-25562R2

Dear Dr. Fite,

We’re pleased to inform you that your manuscript has been judged scientifically suitable for publication and will be formally accepted for publication once it meets all outstanding technical requirements.

Kind regards,

Frank T. Spradley

Academic Editor

PLOS ONE

---

## [Editor Report · Acceptance letter]

31 Mar 2021

PONE-D-20-25562R2 

Compliance with Iron and folic acid supplementation (IFAS) and associated factors among pregnant women in Sub-Saharan Africa:  A systematic review and meta-analysis 

Dear Dr. Fite:

I'm pleased to inform you that your manuscript has been deemed suitable for publication in PLOS ONE. Congratulations! Your manuscript is now with our production department. 

Kind regards, 

on behalf of

Dr. Frank T. Spradley 

Academic Editor

PLOS ONE